# An Adversarial Example for Direct Logit Attribution: Memory Management in GELU-4L

**Jett Janiak**[*]
Independent

**Can Rager**[*]
Independent

**James Dao**[*]
Independent

**Yeu-Tong Lau**[*]
Independent

## Abstract

Prior work suggests that language models manage the limited bandwidth of the residual stream through a "memory management" mechanism, where certain attention heads and MLP layers clear residual stream directions set by earlier layers. Our study provides concrete evidence for this erasure phenomenon in a 4-layer transformer, identifying heads that consistently remove the output of earlier heads. We further demonstrate that direct logit attribution (DLA), a common technique for interpreting the output of intermediate transformer layers, can show misleading results by not accounting for erasure.

## 1 Introduction

Understanding the internal mechanisms of language models is an increasingly urgent scientific and practical challenge (Zhao et al., 2023; Luo and Specia, 2024). For instance, we lack a clear explanation of the interaction between internal components, such as attention heads and MLPs. Elhage et al. (2021) referred to residual stream dimensions as *memory* or *bandwidth* that components use to communicate with each other.

**Memory management** Elhage et al. (2021) observe that there are much more computational dimensions (such as neurons and attention head result dimensions) than residual stream dimensions, thus we should expect residual stream bandwidth to be in high demand. The authors speculated that some model components perform a *memory management* role, clearing residual stream dimensions set by earlier components to free some of this bandwidth.

**Direct logit attribution (DLA)** is a technique for interpreting the output activations of model components in vocabulary space (Wang et al., 2022; Elhage et al., 2021; nostalgebraist, 2020). In particular, DLA applies the unembedding matrix to

---

[*] Equal contribution. Correspondence to jettjaniak@gmail.com

model internal activations, effectively skipping further computation of downstream components. This method implicitly assumes continuity of the residual stream, meaning a direction written to the stream stays conserved throughout the forward pass. However, the continuity assumption would not hold if some components erase residual directions set by earlier ones. Overall, our main contributions are as follows:

- Defining *erasure*, a form of memory management in transformer models and proposing *projection ratio*, a metric for quantifying erasure

- Presenting a concrete example of erasure in a 4-layer transformer

- Demonstrating that DLA can yield misleading results when erasure is present

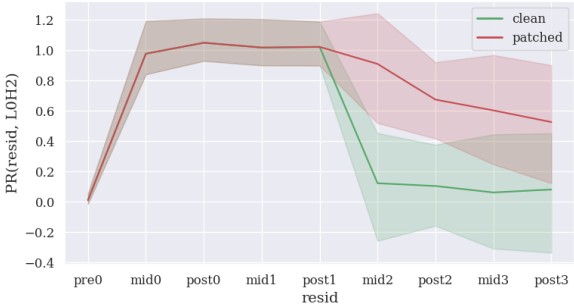

Figure 1: The output of attention head L0H2 across the residual stream with (green) and without (red) erasure behavior. We show the median projection ratios between residual stream activations and L0H2, with and without V-composition patching. Shaded region represents 25th and 75th quantiles.

## 2 Methods

We characterize *erasure* as 3 steps during a forward pass of a model: (1) A *writing component*

adds its output to the residual stream. (2) Subsequent components read this information to perform their function. (3) An *erasing component* removes the writing component's output from the residual stream, by reading it and writing out a negative version.

## 2.1 Identifying writing components

We examine whether the output of each component, once written to the residual stream, persists in subsequent transformer layers. To quantify this, we define the *projection ratio*

$$\text{PR}(\mathbf{a}, \mathbf{b}) := \frac{\mathbf{a} \cdot \mathbf{b}}{||\mathbf{b}||^2}, \tag{1}$$

which measures the proportion of vector $\mathbf{b}$ present in vector $\mathbf{a}$. We set $\mathbf{a}$ to be the residual stream activations at each layer and $\mathbf{b}$ to be the output of each attention head or MLP. This allows us to track how much of each component's output remains in the residual stream as it propagates through the model.

## 2.2 Identifying erasing components

To identify erasing components, we look for components that write to the residual stream in the direction opposite to the previously identified writing components. We quantify this with the projection ratio, this time setting $\mathbf{a}$ to be the output of a writing component and $\mathbf{b}$ outputs of other components.

## 2.3 Investigating causality

To investigate a causal relationship between writing and erasure, we repeat experiments identifying writing and erasing components, while intervening on the direct path between them with activation patching (Zhang and Nanda, 2023).

Specifically, to compute the value vector of an erasing attention head, we use a modified residual stream activation, where the output of the writing component is set to zero. In other words, we perform activation patching with zero ablation to the V-composition (as defined by Elhage et al. (2021) and applied by Wang et al. (2022); Heimersheim and Janiak (2023); Lieberum et al. (2023)) of writer and erasure heads. Put simply, V-composition is the direct path between the output of an upstream component and the value input of a downstream attention head.

Zero-ablation of the writing component's output allows us to observe the impact on the erasing behavior and establish a causal link between the two

components. For example, to investigate the causality of L0H2 (an early writing component) on L2H2 (a later erasing component), we can subtract the output of L0H2 from the value input of L2H2. This helps answer the question "how does L2H2 behave differently when L0H2's output is not present?".

## 2.4 Erasure as a potential confounder in DLA interpretation

We hypothesize that erasure can lead to misleading results when using DLA to interpret the role of writing components. If an erasing component removes the output of a writing component from the residual stream, the writing component's contribution to the final logits (as measured by DLA) will be diminished, as the effects of the two components will largely cancel out.

To test this, we collect prompts from the model's training dataset and measure the contribution of the identified writing and erasing components to the logit difference between the model's top two next token predictions. We isolate the erasing effect by applying DLA only to the part of the erasing components' output that comes from V-composition with the writing component. This is obtained by taking the erasing components' output on a standard forward pass and subtracting their output from a modified forward pass where the writing component's output is zeroed out in the residual stream.

## 2.5 Verifying DLA predictions through context manipulation

To find examples that yield significant DLA results for the writing component, we search for tokens whose unembedding directions consistently align with the writing component's output. Having identified tokens that yield significant DLA results, we investigate whether these results are genuine contributions of the writing component or artifacts of erasure. For each selected token, we construct a prompt that makes the token a natural next-word prediction, and the model indeed predicts it as the most likely continuation.

We then measure the logit difference between the selected token and the model's second most likely prediction using DLA in two scenarios: (1) a clean run with the original prompt and (2) a run where the input to the writing component is patched with randomly sampled prompts from the training dataset. If the writing component is genuinely using the information in the prompt to infer the best prediction, then patching its input should signifi-

cantly reduce the logit difference observed in the clean run. Conversely, if the DLA predictions are primarily artifacts of erasure, patching the input should have little impact on the observed logit differences.

## 2.6 Model architecture and training

For our experiments, we utilized a GELU-4L model (Nanda, 2022). This model is based on a GPT-2 style transformer architecture with 4 transformer layers, learned positional embeddings, and layer normalization. It employs GELU activations in the MLP layers, uses separate embedding and unembedding matrices (not tied), and has a residual stream dimension of 512. The model was trained on a dataset of 22 billion tokens, comprising 80% web text and 20% Python code.

## 3 Results

### 3.1 Output of head L0H2 is being erased

We measured the projection ratio between residual stream activations at subsequent layers and outputs of every transformer component in forward passes on 300 random samples of the model's training data.

We distinguish the states of the residual stream in GELU-4L as follows: resid_pre_0 before any attention or MLP layers (just token and positional embeddings), resid_mid_$n$ after the attention layer $n$, and resid_post_$n$ after the MLP layer $n$, where $n = 0, 1, 2, 3$ denotes the layer index.

The most interesting results were observed for attention head 2 in layer 0 (L0H2), shown by the green line (clean) in Figure 1. We can track the presence of L0H2's information in the residual stream across subsequent layers of the model.

Initially, we see a projection ratio close to 0 at resid_pre_0, as L0H2 has not written to the residual stream yet. After L0H2 writes to the residual stream at resid_mid_0, the projection ratio goes to about 1, meaning its output is fully present in the residual stream. The projection ratio stays close to 1 between resid_mid_0 and resid_post_1. However, between resid_post_1 and resid_mid_2, attention heads appear to remove the information that L0H2 originally wrote, resulting in a much smaller projection ratio, close to 0.

### 3.2 Layer 2 attention heads are erasing L0H2

In Figure 2, we can see the projection ratio between the outputs of every component in layers 1

to-3 and the output of head L0H2. We find that 6 out of 8 attention heads in layer 2, numbered 2 to 7, have consistently negative projection ratio, implying that they are writing to the residual stream in the direction opposite to L0H2. In aggregate, they are responsible for erasing 90.7% [1] of the output of L0H2. We refer to them as *erasing heads*.

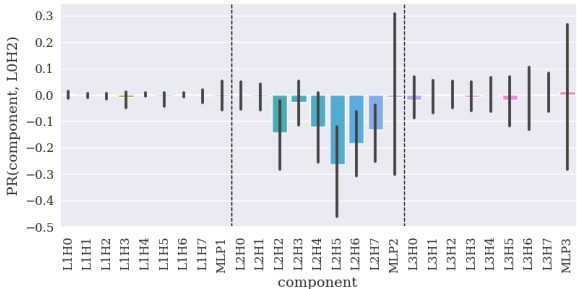

Figure 2: Median projection ratios between components in layers 1–3 and head L0H2. Error bars represent 25th and 75th quantiles.

### 3.3 Erasure depends on writing

Figure 1 shows the projection ratio of residual stream onto L0H2 in the clean run and in a patched run, where we prevented V-composition between L0H2 and erasing heads. As we can see, in the patched run the projection ratio remains high after the attention block in layer 2 (0.91 in patched, 0.12 in clean), indicating that around 85% of the erasure in layer 2 is dependent on V-composition. We note that the projection ratio goes down after layer 2, suggesting that components in subsequent layers are involved in the erasure as well.

Figure 3 compares projection ratios between erasing heads and L0H2 in patched and clean runs. While these heads express consistently negative projection ratios in the clean run, the median goes close to zero in the patched run. These results show that the erasure behavior disappears when we prevent V-composition between L0H2 and the erasing heads.

### 3.4 DLA contributions of writing and erasure are highly anti-correlated

To investigate how erasure can affect the interpretation of writing components using DLA, we applied the methodology described in Section 2.4. We collected 30 random samples from the model's

---

[1]The distribution of projection ratio between the sum of erasing heads output and L0H2 has quantiles: 25% = -1.128, 50%=-0.907, 75%=-0.700.

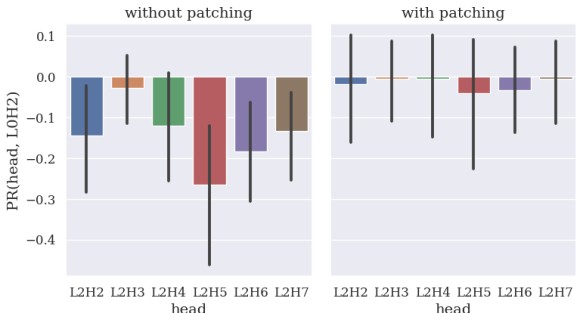

Figure 3: Median projection ratios between selected heads in layer 2 and head L0H2, with and without V-composition patching. Error bars represent 25th and 75th quantiles.

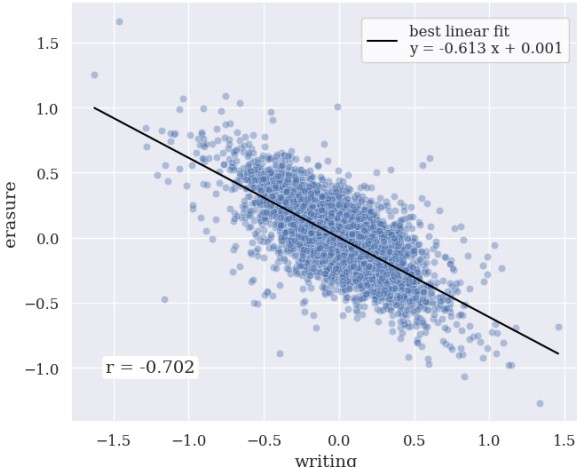

Figure 4: Correlation between the effects of writing and erasure on the logit difference of top 2 model predictions, according to DLA.

training dataset and considered the top 2 next token predictions at every sequence position.

The results, shown in Figure 4, reveal a strong negative correlation (r=-0.702) between the DLA contributions of the writing head L0H2 and the erasing heads in layer 2. The line of best fit has a slope of -0.613, indicating that on average, the erasing heads remove about 61% of L0H2's apparent contribution to the final logits, as measured by DLA.

This anti-correlation suggests that DLA results for the writing component L0H2 may be largely artifacts of the downstream erasure. When the writing component appears to make a large contribution to the final logits according to DLA, the erasing components tend to make a similarly large contribution in the opposite direction. As a result, the net effect of the writing component on the final output may be much smaller than what DLA alone would suggest.

### 3.5 Adversarial examples of high DLA values without direct effect

We selected four tokens for which the unembedding direction aligns with the output of L0H2: `" bottom"`, `" State"`, `" __"`, and `" Church"`. Then, we constructed four prompts such that the model predicts one of the tokens with highest probability.

1. prompt: `"It's in the cupboard, either on the top or on the"`
   top-2 tokens: `" bottom"`, `" top"`
   (logit difference 1.07)

2. prompt: `"I went to university at Michigan"`

top-2 tokens: `" State"`, `" University"`
(logit difference 1.89)

3. prompt: `"class MyClass:\n\tdef"`
   top-2 tokens: `" __"`, `" get"`
   (logit difference 3.02)

4. prompt: `"The church I go to is the Seventh-day Adventist"`
   top-2 tokens: `" Church"`, `" church"`
   (logit difference 0.94)

We use the methodology described in Section 2.5. We find that patching the input to L0H2 with unrelated text does not affect the DLA-measured logit difference, as shown in Figure 5 (top). Therefore, we conclude that L0H2 does not directly contribute to the model predictions in prompts 1 to 4, despite significant DLA values.

For example, if we change Prompt 1 to a context completely different to the vertical placement of an object in a cupboard (such as in the patched run), we no longer expect the model to differentially boost the logit of `" bottom"` over `" top"`. However, DLA of L0H2 still suggests that L0H2 is indeed differentially boosting the `" bottom"` token, and this remains true for 300 randomly sampled inputs.

The invariance of L0H2's DLA to input tokens is unusual. We reran the patching experiment for four other attention heads that, according to DLA, have the highest direct effect on logit difference for the respective prompt in Figure 5 (bottom). In

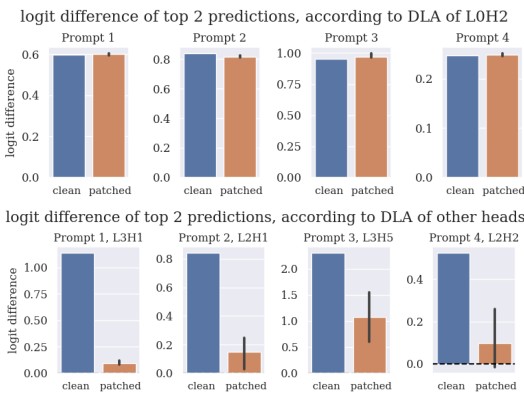

Figure 5: Logit difference of top 2 predictions on adversarial examples, according to DLA. Patched refers to replacing the input to the head L0H2 (top) or other heads with high logit difference according to DLA (bottom) with one from a run on unrelated text with the same number of tokens (300 examples). The orange bars show median with error bars at the 25th and 75th quantiles.

contrast to L0H2, the results for these heads are severely affected by the patch, as expected.

## 4 Conclusion

In this paper, we presented a concrete example of memory management in a 4-layer transformer model. It is important to note that our study focused on a single model and a specific attention head. Further research is needed to determine the extent to which these phenomena generalize across different model components and model sizes.

Our findings also highlight the need for caution when using DLA, as in the presence of the erasure phenomenon, these results can be misleading. To mitigate this, we advocate for testing effects across varied prompts, particularly those with different correct next token completions, as averaging over many prompts can cancel out spurious results. Moreover, we recommend complementing DLA with activation patching to measure both direct and indirect effects of model components.

## Acknowledgments

Our research benefited from discussions, feedback, and support from many people, including Chris Mathwin, Evan Hockings, Neel Nanda, Lucia Quirke, Jacek Karwowski, Callum McDougall, Joseph Bloom, Sam Marks, Aaron Mueller, Alan Cooney, Arthur Conmy, Matthias Dellago, Eric Purdy, and Stefan Heimersheim.

Part of this work was produced during ARENA 2.0 and MATS Program – Spring 2023 Cohort.

We conducted our experiments using the GELU-4L model trained by Neel Nanda (Nanda, 2022) and the TransformerLens library (Nanda and Bloom, 2022), which were instrumental in our research.

## Impact Statement

This paper presents work whose goal is to advance the field of Machine Learning. There are many potential societal consequences of our work, none which we feel must be specifically highlighted here.

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
