# OpenReview forum: "An Adversarial Example for Direct Logit Attribution: Memory Management in GELU-4L"
_EMNLP/2024/Workshop/BlackBoxNLP — BlackboxNLP 2024_

### Official Review · Reviewer_jSkP · 2024-09-09

**Overall Assessment:** 3
**Confidence:** 4

**Best Paper:**

1

**Best Paper Justification:**

n/a

**Comments Questions Suggestions And Typos:**

na

**Paper Summary:**

The authors show that transformer models erase earlier layer outputs and suggest that direct logit attribution (DLA) can give misleading interpretations.

**Summary Of Strengths:**

This paper illustrates memory management in transformer models, through residual stream erasure. The authors also identify attention heads that consistently remove earlier outputs, contributing to a deeper understanding of transformer mechanics. The study also highlights the potential for misinterpretations when using Direct Logit Attribution without accounting for this erasure effect. To address these issues, the authors  also offer recommendations, such as averaging results over varied prompts and using activation patching to better measure both direct and indirect effects of model components.

**Summary Of Weaknesses:**

The study uses a single 4-layer transformer model, that may limit the generalizability of its findings to larger or more complex models.

---

### Official Review · Reviewer_1Yeq · 2024-09-11

**Overall Assessment:** 2
**Confidence:** 2

**Best Paper:**

1

**Best Paper Justification:**

N/A

**Comments Questions Suggestions And Typos:**

In standard Transformer architecture, the output of heads is concatenated across each layer, meaning that the output of a specific head has a different dimensionality than the residual channel. How do you compute the projection ratio in such a case?

**Paper Summary:**

The work aims to study the interactions between distinct components of a tiny transformer model (a 4-layer model based on GPT-2 architecture). Specifically, one head in the first layer (L2H0) is selected as the so-called "writing component"; then, the analysis identifies "erasing components" as the ones with the output vectors opposite to the output vector of L2H0. Such "erasing components" are found mainly in the 3rd layer of self-attention. The findings are interpreted as a counter-example for assumptions made in the direct logit attribution method, which in previous literature was used to quantify the effect of model components on the predictions.

**Summary Of Strengths:**

- The raised research problems of interactions between components' output are interesting and deserve more research.

- The work discovers unknown limitations of a direct logit attribution method that was used as an interpretability tool in previous research works.

**Summary Of Weaknesses:**

- The scope of the work should be extended to other attention heads and components (ideally models) to check if the observed pattern is repeated anywhere or if it is just a feature of the specific L0H2 head.

- The experimental design choices are not adequately justified; the authors should explain why they picked the tiny Transformer model, the specific L0H2 head, and the prompt (some of which are ungrammatical).

Authors should highlight the meaningfulness of their findings. From the results, it appears that even the highest projection ratios are relatively low, and "erasing heads" would not erase the whole information "written" by L0H2. Perhaps "direct logit attribution" and its limitations should be analyzed in more detail in the experimental part.

- The writing could be more coherent. The definitions of the introduced concepts are too condensed and often hard to comprehend.

---

### Decision · Program_Chairs · 2024-09-19

**Decision:**

Accept

**Comment:**

The reviewers agreed that the topic and findings of the paper was interesting and would inspire future work. The reviewers, however, also pointed out that there are some arbitrary and narrow modeling decisions that may limit the generalizability of the findings. Nevertheless, presenting this work at BlackboxNLP may help develop/refine this project further in a way that addresses the reviewers' concerns.